# Ocular Surface Characteristics in Pugs with Pigmentary Keratitis in the Canary Islands, Spain

**DOI:** 10.3390/ani14040580

**Published:** 2024-02-09

**Authors:** Diana Sarmiento Quintana, Inmaculada Morales Fariña, Jéssica González Pérez, José Raduan Jaber, Juan Alberto Corbera

**Affiliations:** 1Hospital Clínico Veterinario, Facultad de Veterinaria, Universidad de Las Palmas de Gran Canaria, 35413 Las Palmas, Spain; 2Dioftalmo, Oftalmología Veterinaria, 35003 Las Palmas, Spain; 3Instituto Universitario de Investigaciones Biomédicas y Sanitarias (IUIBS), Universidad de Las Palmas de Gran Canaria, 35016 Las Palmas, Spain; 4Clínica Veterinaria Escaleritas, Las Palmas de Gran Canaria, 35011 Las Palmas, Spain; 5Departamento de Morfología, Universidad de Las Palmas de Gran Canaria, 35016 Las Palmas, Spain

**Keywords:** pigmentary keratitis, Pug-breed dogs, corneal thickness, ocular diseases, corneal vascularization, tear film

## Abstract

**Simple Summary:**

The problem addressed in this study is the development of pigmentary keratitis (PK) in Pug-breed dogs, a condition characterized by corneal pigmentation and associated with ocular manifestations and vision loss. The aim of the study was to assess the occurrence of PK in Pug dogs, identify predisposing factors, and evaluate the tear film quantity and quality in affected dogs. The study found that the prevalence of PK was high in dogs, with 94.5% of eyes showing PK. The severity of corneal pigmentation varied, with very mild (20.3%), mild (16.4%), moderate (38.2%), and severe (25.1%) categories observed. Age and previous ocular diseases were identified as potential risk factors for the development of severe PK. The main conclusion of the study is that PK in Pug-breed dogs in the Canary Islands has a high prevalence and may have a genetic basis. Our most relevant contribution is the evaluation based on the Tear Break-Up Time (TBUT) test and the Tear Ferning Test (TFT). This information can be valuable to society by helping veterinarians diagnose and treat PK in Pugs and potentially prevent the disease through genetic screening and breeding practices.

**Abstract:**

This study investigated the prevalence of pigmentary keratitis (PK) in Pug-breed dogs and described the ocular surface characteristics associated with this disease. A total of 219 eyes from 110 dogs were examined, with 94.5% of them affected by PK. Age, previous ocular diseases, corneal vascularization, and corneal sensitivity were significantly associated with the presence of PF and the severity of corneal pigmentation. The study also found that low tear production and blinks incomplete with tear signs, as well as reduced corneal sensitivity, were linked to more severe forms of corneal pigmentation. The Tear Ferning Test (TFT) was identified as a valuable tool for evaluating tear quality in dogs, with worse test results indicating a higher risk of severe PK. A lower mean Tear Break-Up Time (TBUT) test was observed in dogs with PK. Additionally, the study observed a statistically significant difference in corneal thickness between the nasal and temporal zones, with the nasal zone being thicker. It was also suggested that sex and fertility status may influence the incidence of PK and the severity of corneal pigmentation. Overall, these findings provide insight into the underlying causes of PK in Pugs and can inform future treatment strategies for this breed.

## 1. Introduction

Corneal pigmentation is a biological response that occurs in the cornea due to various irritating stimuli, including mechanical, immune-mediated, and tear film disorders such as keratoconjunctivitis sicca [1,2,3]. It has been observed that brachycephalic breeds, including the Pug, Shih Tzu, Lhasa Apso, Pekingese, and Boston Terrier, are particularly susceptible to rapid and pronounced corneal pigmentation [1,4,5,6,7]. Furthermore, corneal pigmentation has been associated with immunological disorders such as chronic superficial keratitis (CSK) or pigmentary pannus, keratoconjunctivitis sicca (KCS), chronic ulcerative keratitis, and pigmentary keratitis syndrome in brachycephalic breeds [4,5,8,9,10,11].

Pigmentary keratitis (PK) is a clinical condition observed commonly in brachycephalic dogs, primarily due to chronic irritation of the cornea [1,5,12,13]. The development of localized corneal pigmentation usually initiates at the nasal cornea and expands throughout the ocular surface [14]. This pigmentation is hypothesized to arise from the infiltration of pigmented corneal epithelial injury, which surpasses the regenerative capability of the corneal epithelial cells [15]. More recently, Vallone et al. [8] presented microscopic evidence of corneal melanin accompanied by inflammatory pathology, further supporting the adoption of the term “pigmentary keratitis” to describe this condition. The presence of pigment on the cornea can lead to visual impairment and, in severe cases, blindness [16].

Recent studies have reported a high prevalence of PK in Pug dogs [1,4,5,8,17]. However, the exact cause of this condition is still not fully understood, and there is a lack of defined treatment strategies. Moreover, the severity of the disease seems to be linked to the presence of melanosis in the corneal limbus and the density of corneal melanosis, suggesting a predisposition in this breed [18,19].

The evaluation of tear quantity can be conducted utilizing both the Schirmer Tear Test (STT) and the phenol red thread test. Additionally, the evaluation of tear quality can be performed through the Tear Break-Up Time (TBUT) test and the Tear Ferning Test (TFT) [20,21,22].

The objectives of this research were to assess the prevalence of PK in Pug-breed dogs within our geographical setting, to examine the predisposing factors contributing to PK in this cohort, and to investigate the quantitative and qualitative evaluation of the tear film in Pug-breed dogs affected by PK and its relationship to the severity of corneal pigmentation characteristic of the corneal surface.

## 2. Materials and Methods

A prospective clinical study was undertaken at the Veterinary Teaching Hospital of the Veterinary Faculty of the University of Las Palmas de Gran Canaria (ULPGC) in the Canary Islands, Spain. To gather epidemiological data, personal interviews were conducted with the owners or caretakers (Appendix A).

Data on age, sex, fertility status, and coat color were gathered from the participants. The coat color was classified according to the breed standard established by the Fédération Cynologique Internationale (FCI) and divided into four categories: silver, apricot, fawn, and black. To examine the potential impact of coat color, we grouped them into two categories: light coats (silver, apricot, and fawn) and dark coats (black). Furthermore, a detailed medical history was obtained, including any prior ocular ailments, treatments (such as the application of immune-suppressing topical agents or eyelid surgeries), and any systemic diseases.

### 2.1. Animals

A complete clinical and ophthalmological examination was conducted on a cohort of 110 Pugs, encompassing a total of 219 eyes. The age range of the canine subjects ranged from 1 to 13 years, with a mean age of 6.6 ± 2.5 years. Among the participants, 51 individuals were female (46.4%) and 59 were male (53.6%). Additionally, the fertility status of each animal was documented, revealing that 33 subjects were neutered (30%) while the remaining 77 remained intact (70%). Notably, one subject had undergone enucleation due to a traumatic ocular injury, which was unrelated to the present study.

### 2.2. Ophthalmologic Examination

Ocular images were acquired utilizing a Canon EOS 550 D camera (Canon España S.A., Madrid, Spain) fitted with a 58 mm lens. Magnification during examination was achieved through direct visualization and the utilization of a Kowa SL-15™ slit-lamp biomicroscope (Kowa Europe GmbH, Düsseldorf, Germany). Palpebral modifications that were observed involved euryblepharon, trichiasis of the caruncle, trichiasis of the nasal fold, distichiasis, as well as entropion of the upper and lower nasal regions.

Corneal pigmentation was categorized in accordance with the classification system described by Labelle et al. [1]. Pigmentation was regarded as very mild (1) when it affected less than 2 mm of the corneal limbus, mild (2) when it involved less than 20% of the corneal surface, moderate (3) when it affected 20–50% of the corneal surface, and severe (4) when it affected more than 50% of the corneal surface. Additionally, the presence of corneal vascularization was documented for each eye.

Iris hypoplasia and persistent pupillary membranes (PPMs) were classified into four distinct types following the guidelines proposed by Gelatt et al. [11]. These types are as follows: Type I, characterized by the presence of pigment in the crystalline lens; Type II, where the membrane extends from iris to iris; Type III, where the membrane extends from the iris to the cornea; and Type IV, where the membrane extends from the iris to the crystalline lens.

### 2.3. Tear Film Analysis

To quantitatively evaluate the tear film, we utilized the Schirmer Tear Test-1 (STT) Eickemeyer™, (Eickemeyer Veterinary Technology for Life Inc., Stratford, Ontario, Canada). Qualitative assessment, on the other hand, was performed using the TBUT test, Fluorescein Sodium Ophthalmic Strips U.S.P^TM^ (Prótesis Hospitalarias S.A, Madrid, Spain), and the TFT. For the TFT, we collected tear samples from the lower conjunctival sac of both eyes using micro-hematocrit tubes (made of neutral glass by Vitrex Medical A/S, Herlev, Denmark). These tear samples were then allowed to dry on glass slides (provided by Knittel Glass, Braunschweig, Germany) for a duration of 10 min at a temperature range of 21–23 °C and relative humidity of 60–80%. Subsequently, we observed the slides under an optical microscope (Olympus CH20, Olympus Iberia S.A.U., Barcelona, Spain) at a magnification of 10X To categorize the fern patterns, we employed a grading scale developed by Williams and Hewitt [23], ranging from abundant and closely spaced ferns (grade I) to the absence of ferns (grade IV). Grades I and II were categorized as good quality-normal, while grades III and IV were classified as poor quality-abnormal (Figure 1).

Following the TBUT test and TFT, the eyes were subsequently rinsed with a saline solution and examined for the fixation of fluorescein onto the cornea. Corneal sensitivity was assessed using qualitative esthesiometry. To minimize variations in the methodology, all measurements were performed by a single observer. Corneal fine sensitivity was determined using a 0.2 mm filament made of monosyn 4/0 material, while gross sensitivity was assessed using a rough 1 mm cotton wool. Three tactile stimulations were executed, and the resulting reaction was categorized as negative, reduced (if a corneal blink occurred after 3–5 stimulations), or positive (indicated by a rapid blink reflex).

The corneal thickness was assessed using an ultrasound pachymeter, specifically the Pachette 3 ultrasonic pachymeter from DGH Technology Inc. (Exton, PA, USA) Prior to measurement, oxibuprocaine hydrochloride and tetracaine hydrochloride (Colircusi^TM^ double anesthetic, 1 mg/mL + 4 mg/mL from Alcon Cusí, Barcelona, Spain) were topically applied for anesthesia. A total of 25 measurements were obtained from both the temporal and nasal regions of the cornea, and the mean and standard deviation were calculated.

### 2.4. Statistical Analysis

The statistical analysis was conducted using IBM SPSS Statistics version 23 for Windows. A significance level of *p* < 0.05 was considered to indicate statistical significance. Descriptive statistics were calculated for each parameter, including frequency and valid percentages, as well as mean, standard deviation, maximum, minimum, and sample size (n) for the eyes or animals included in the study. For the comparative statistical analysis, Pearson’s Chi-square test (χ^2^) was used when comparing two qualitative variables. If the count was less than 5, Fisher’s exact test was utilized.

An analysis of variance (ANOVA) was employed to investigate one categorical and one numerical variable. Levene’s test was utilized to evaluate the equality of variances. Non-parametric tests for independent samples, namely the Mann–Whitney U hypothesis test and the Kruskal–Wallis test, were employed to determine whether to accept or reject the null hypothesis when the results were found to be statistically significant.

We examined the predisposing factors and evaluated their statistical significance in relation to the occurrence of severe corneal pigmentation. This was carried out by calculating the odds ratio (OR) to compare the group with severe corneal pigmentation to the other groups. The distribution of certain parameters did not follow a normal pattern, as indicated by the Kolmogorov–Smirnov test. Therefore, the Friedman Test was used for qualitative variables in the efficacy study, while the Wilcoxon Test was used for quantitative variables.

## 3. Results

### 3.1. Prevalence of PK

A total of 110 dogs were included in the study. Out of these 219 eyes, 207 (94.5%) were observed to have PK, while the remaining 12 (5.5%) were healthy. The prevalence of PK was found to be the same in both the right eye (OD) and left eye (OS), with 104 out of 110 dogs having PK in their right eye and 103 out of 110 dogs having PK in their left eye.

The severity of corneal pigmentation was categorized into four groups: very mild (20.3%; 42 out of 207 eyes), mild (16.4%; 34 out of 207 eyes), moderate (38.2%; 79 out of 207 eyes), and severe (25.1%; 52 out of 207 eyes), according to the criteria proposed by Labelle et al. [1]. In Figure 2, an example of the four groups is shown.

### 3.2. Previous Ocular Diseases

Out of the 219 eyes that were part of the study, 94 (42.9%) had a documented history of ocular diseases, while the remaining 125 (57.1%) were reported to have no ocular pathologies, as reported by their owners.

The largest proportion of the study population, comprising 93 animals (84.5%), had not received any topical treatment with immunosuppressants in the six months prior to the study. However, 17 animals (15.5%) had been treated with topical immunosuppressants (cyclosporine or tacrolimus) within six months prior to the study. Furthermore, 16 animals (14.5%) had undergone palpebral alterations surgery, specifically nasal canthoplasty.

Significant statistical differences were noted when comparing the various severity groups and the occurrence of a preexisting ocular condition (*p* = 0.014). Additionally, a prior ocular disease was determined to be linked to an increased risk of developing severe PK (OR = 2.400; 95% CI: 1.260–4.572).

All animals included in this study (n = 219) exhibited pigment in the sclera, with a more pronounced presence in animals with PK. Euryblepharon and trichiasis of the caruncle were present in all animals except for those that had undergone surgical treatment (n = 32 eyes). Inferior nasal entropion was observed in 217 animals (99.1%), while trichiasis of the nasal fold, distichiasis, and upper nasal entropion were found in 16 (7.3%), 28 (12.8%), and 34 (15.5%) animals, respectively. Furthermore, iris hypoplasia was observed in 46.6% of the eyes, and persistent pupillary membranes of type II (extending from iris to iris) were observed in 57.5% of the eyes.

No statistically significant differences were found in palpebral alterations, except for distichiae. There were no statistically significant differences between the severe group and the other groups in terms of palpebral alterations (χ^2^ = 1.498; *p* = 0.221). There were also no statistically significant differences in the detection of iris hypoplasia (*p* = 0.366) and persistent pupillary membranes (*p* = 0.067). Even when comparing the severe group to the other groups, there were no statistically significant differences in the detection of iris hypoplasia (*p* = 0.145) or persistent pupillary membranes (*p* = 0.529). Detailed information on previous ocular diseases has been included in Table 1.

Corneal vascularization was observed in 57.9% of the eyes examined (Table 1). Statistically significant differences (*p* < 0.001) were observed in corneal vascularization among different levels of corneal pigmentation severity. The severe group exhibited significant differences (χ^2^ = 33.605; *p* < 0.001) compared to the other groups. An increasing severity of corneal pigmentation was associated with a higher risk of corneal vascularization presence (OR = 13.833, 95% confidence interval: 4.756–40.238).

### 3.3. Predisposing Factors

The predisposing factors contributing to the development of PK were examined by assessing multiple variables, as delineated in Table 2.

#### 3.3.1. Age

A statistically significant difference in the prevalence of PK in the eyes was observed among different age groups (Table 2). Animals aged 8 years or older had a significantly increased risk of PK (OR = 16.676; 95% CI: 3.183–87.372) compared to the youngest group (0 to 3 years), as did animals aged 4 to 7 years (OR: 14.961; 95% CI: 3.524–63.515). However, no significant differences (*p* = 0.906) were found between the group of animals aged 4 to 7 years and the group aged 8 years or older.

We further calculated the odds ratio to quantify the risk associated with age and the severity of corneal pigmentation compared to the rest of the cases. The calculated OR was 4.340 with a 95% confidence interval (CI) of 2.230–8.449. Thus, our findings suggest that older animals are at a higher risk of developing a more severe form of PK (Table 3).

#### 3.3.2. Sex and Fertility Status

No significant differences were observed between males (113/118; 95.8%) and females (94/101; 93.1%) (*p* = 0.383) or between fertility status and the presence or absence of PK (*p* = 0.690) (Table 2). The analysis conducted to assess the relationship between sex, fertility status, and the severity of corneal pigmentation did not yield any statistically significant differences (*p* = 0.765 and *p* = 0.103, respectively) (Table 3). Additionally, when comparing the severe corneal pigmentation group with regards to sex and fertility status, no significant differences were observed (χ^2^ = 0.590; *p* = 0.442 and χ^2^ = 1.222; *p* = 0.269, respectively).

#### 3.3.3. Coat Color

No significant differences were identified when comparing color coats, which were categorized as dark coats (30/32; 93.7%) or light coats (177/187; 94.6%), and the presence of PK (*p* = 0.836) (Table 2). Furthermore, no correlation was found between the dark and light color coat groups and the severe corneal pigmentation group (χ^2^ = 1.333; *p* = 0.248) (Table 3). These findings suggest that coat color does not represent a risk factor for the development of severe PK.

### 3.4. Tear Film Analysis

#### 3.4.1. Schirmer Tear Test (STT)

Quantitative analysis of tear film demonstrated a mean STT value of 16.3 ± 5.8 mm/min. The Schirmer Tear Test values of patients were compared with and without PK. Our findings demonstrate that there is no statistically significant difference between these two groups (*p* = 0.102). Among participants with PK, the STT values were found to be 16.11 ± 5.85 mm/min (N = 207), while participants without PK had values of 18.92 ± 3.78 mm/min (N = 12). In terms of tear film measurements, there was no statistically significant difference (*p* = 0.408) between eyes with PK and those without (Table 4). A statistically significant difference (*p* < 0.001) was observed upon comparison of the outcomes obtained from STT and the severity of corneal pigmentation. The most diminished value for STT was observed in cases with the highest severity of corneal pigmentation. However, the Mann–Kendall trend test did not provide conclusive evidence of a diminishing trend (*p* = 0.083) (Table 5).

#### 3.4.2. Tear Ferning Test (TFT)

The TFT results, evaluated using Rolando’s scale [23], were categorized as type I (12.8%), type II (38.5%), type III (29.6%), and type IV (19%). The eyes without PK did not exhibit the lowest tear quality, as evidenced by the TFT type IV (0/34; 0%). More detailed information can be found in Table 4. In our study, no correlation was observed between the results of the TFT and the detection of PK. However, we did find significant differences when comparing corneal pigmentation severity groups (*p* = 0.002) (Table 5).

The results of the TFT were categorized into two groups: a low-quality abnormal group (type III and type IV) and a good-quality normal group (type I and type II). We observed a statistical difference between the severe group and the rest (χ^2^ = 11.515; *p* = 0.001). Furthermore, we found that the worse the TFT result, the higher the risk of severity in corneal pigmentation (OR = 3.540, 95% CI: (1.667–7.517)).

#### 3.4.3. Tear Break-Up Time (TBUT)

The average TBUT test was measured to be 6.0 ± 2.7 s (Table 4). Following the study of the relationship with the presence of PK, a statistically significant difference was observed in the TBUT test (*p* = 0.038). Eyes without PK exhibited a higher mean TBUT test (7.6 ± 2.2 s; N = 12) compared to those with PK (5.9 ± 2.7 s; N = 207) (Table 3).

On the other hand, the groups with different severity of corneal pigmentation showed statistically significant differences (*p* < 0.001) when comparing TBUT test scores. The mean TBUT test scores decreased as the severity increased. Furthermore, no discernible trend was observed in the series (*p* = 0.083) (Table 5).

### 3.5. Corneal Sensitivity

Fine sensitivity was observed to be negative in 148 eyes (67.6%) or reduced in 71 eyes (32.4%). Gross sensitivity was found to be positive in only 22 eyes (10%), reduced in 175 eyes (79.9%), and negative in the remaining eyes (Table 4). However, no statistically significant differences were found for either fine sensitivity (*p* = 0.945) or gross sensitivity (*p* = 0.397).

There were no statistically significant differences in corneal sensitivity between groups with varying corneal pigmentation severity, either in fine sensitivity (*p* = 0.093) or gross sensitivity (*p* = 0.218) (Table 5). However, a statistically significant difference (*p* = 0.046) was observed when comparing fine corneal sensitivity between the severe corneal pigmentation group and the other groups (OR = 2.108, 95% CI: 1.004–4.427) (Table 5).

### 3.6. Corneal Thickness

Ultrasonic pachymetry analysis yielded a temporal corneal thickness of 697.9 ± 134.1 µm and a nasal corneal thickness of 761.9 ± 144.8 µm. Significant differences in both nasal (*p* = 0.001) and temporal (*p* = 0.014) corneal thickness were observed when comparing the groups with and without PK. Furthermore, the results demonstrate that corneal thickness is increased on the nasal side compared to the temporal side (Table 4).

## 4. Discussion

The etiology of PK remains poorly understood in the scientific literature, necessitating further research for clarification. Some researchers propose that PK is a non-specific biological response to various stimuli, including mechanical abrasion, immune-mediated keratitis, trauma, and tear film disorders [1]. Moreover, the conformation of the eyelids and the quality of the tear film may contribute to the pathogenesis of secondary corneal pigmentation in brachycephalic dogs [5,8,12,24,25,26]. Although previous studies have linked eyelid conformation and its changes to the presence of PK in various dog breeds, Labelle et al. [1] did not observe such a correlation in their study of Pug dogs. These authors did not identify any significant differences in the severity of lagophthalmos or entropion between Pug dogs with and without PK, suggesting that these factors may not be accountable for the development of PK in the Pug breed.

### 4.1. Prevalence of PK

The prevalence of PK has been investigated in Pug-breed dogs in various regions worldwide. High prevalence rates have been reported in Wisconsin and Illinois (USA) at 82.4% [1] and in Austria at 39.1% [17]. Valone et al. [8] described the highest prevalence of Pug dogs at 71.8% in New York (USA), although lower prevalence rates were observed in other brachycephalic breeds (40%). Our findings show a prevalence of 94.5% for PK in Pug-breed dogs in the Canary Islands. Climatic conditions in the Canary Islands could be responsible for this high prevalence. Although several studies detail the impact of environmental conditions on the ocular surface [27], further studies are needed on how the Canary Islands differ from other latitudes, particularly with emerging diseases resulting from climate change. On the other hand, based on our results and those reported in the literature, a genetic basis for PK in Pug-breed dogs cannot be excluded [1].

In our investigation, the majority of cases of PK exhibited a moderate level of corneal pigmentation (38.2%). However, a previous study by Labelle et al. [1] found that the presence of pigment was predominantly categorized as very mild or mild in both the right and left eyes. Additionally, Maggs et al. [18] observed that Pug dogs tended to have more severe corneal pigmentation compared to other breeds. This could potentially be attributed to the association between the melanotic limbus and the density of corneal melanosis. Furthermore, Vallone et al. [8] reported higher severities of corneal pigmentation in Pug dogs, with a mean score of 2.2 in the right eye and 2.1 in the left eye, in contrast to dogs of other breeds with corneal pigmentation, which had mean scores of 1.8 in the right eye and 1.7 in the left eye.

### 4.2. Previous Ocular Diseases

In our study, all animals with a history of ocular disease exhibited PK. The epidemiological survey reported previous ocular surface diseases such as corneal ulcers, infectious conjunctivitis, or dry eye, as well as additional diagnoses made during ophthalmological examination, including euriblepharon, distichiasis, and trichiasis, which may predispose to corneal pigmentation. The exact mechanism underlying corneal pigment accumulation remains unknown, but it is postulated that various factors contribute to this process [28]. According to Wilcock and Njaa [14], corneal pigmentation occurs because of the proliferation of pigmented conjunctival epithelial cells in response to persistent corneal epithelial injury, which cannot be regenerated by resident cells. It is conceivable that prior ocular diseases can cause this persistent injury or irritation, resulting in the accumulation of pigment in the cornea.

Our findings contrast with those of Labelle et al. [1], who did not identify a significant association between a previous ocular disease diagnosis and the presence of PK. However, our findings are in line with those reported by Krecny et al. [17], who observed bilateral euryblepharon and nasal entropion in all Pug-breed dogs included in their study. These findings indicate that these eye conditions are congenital in Pugs.

Our study suggests that palpebral alterations may not be the main cause of PK in Pug-breed dogs. It is possible that the presence of previous ocular diseases contributes to the severity of the PK symptoms, but we cannot conclude that they are part of the underlying cause, as previously suggested by Labelle et al. [1]. These authors proposed that PK in Pugs may have a genetic basis, with the severity of the disease being influenced or exacerbated by factors such as entropion, low tear production, or corneal trauma rather than being solely caused by abnormalities in adnexal conformation or tear quality. Since not all Pug-breed dogs have the same eyelid and facial fold anatomy, it may not be appropriate to apply the same surgical treatment, such as nasal canthoplasty, to correct the condition in all patients. Therefore, it is crucial to individually assess each dog in order to determine the most suitable technique for correcting upper nasal entropion, or nasolabial trichiasis, as described by Andrews et al. [25].

However, Labelle et al. [1] concluded that age is not associated with the presence of PK or the severity of corneal pigmentation, while Krecny et al. [17] found no statistically significant correlation between age and the presence of distichiasis or any other ocular anomaly, including PK.

In this manuscript, we observed that 46.6% of the cases had iris hypoplasia and 57.5% had persistent pupillary membrane (PPM). All PPMs observed were of the iris-to-iris type. These findings support the conclusions of Labelle et al. [1], who found no significant association between the presence of PK or severity of corneal pigmentation and the diagnosis of iris hypoplasia, or PPMs. However, our study reported a lower prevalence of iris hypoplasia (72.1% vs. 46.6%) and PPMs (85.3% vs. 57.5%) compared to Labelle et al. [1], which may be attributed to the higher severity of corneal pigmentation observed in our study, making iris exploration more difficult. Furthermore, we found no statistically significant relationship between iris hypoplasia and PPMs, consistent with Labelle et al. [1]. Only one Pug in the study by Labelle et al. [1] did not display any signs of corneal pigmentation, iris hypoplasia, or PPMs. Additionally, Krecny et al. [17] identified various types of iris alterations in their study, including adherent leukoma, iris-to-iris PPMs, iris-to-crystalline PPM, iris coloboma, and iris atrophy in Pugs. This suggests that these pathologies may be part of an anterior segment dysgenesis syndrome, as hypothesized by Labelle et al. [1]. Furthermore, dyskoria, iris cysts, and iris atrophies were observed in older Pugs.

Corneal vascularization is frequently associated with corneal pigmentation in cases of KCS and CSK, leading to the suggestion that it may serve as the pathway for pigment entry into the cornea [29,30]. However, Labelle et al. [1] reported that approximately half of Pugs with CSK did not exhibit corneal vascularization, suggesting that local cellular pigment production could be responsible for CSK rather than pigment entry through the bloodstream. Krecny et al. [17] investigated 130 Pugs (comprising 258 eyes) and observed corneal pigmentation in 101 eyes, while only 35 eyes exhibited vascularization. Similarly, in our study, 42.0% of eyes affected by PK did not show any corneal vascularization. Nevertheless, we noted that all eyes with corneal vascularization presented with PK. Wilcock and Njaa [16] proposed that mild and transient injuries heal without inflammation due to epithelial slippage and subsequent mitosis of adjacent healthy corneal epithelium. In contrast, more severe and prolonged injuries require the recruitment of conjunctival epithelium, blood vessels, and fibroblasts to facilitate limbal ingrowth and subsequent corneal healing.

Maggs et al. [18] and Wilcock and Njaa [14] have argued against the use of the term “PK” since inflammation is not an inherent component of this reaction. In its place, Labelle et al. [1] have proposed the term “pigmentary keratopathy” for Pug dogs. However, by using a slit-lamp biomicroscope, Krecny et al. [17] and Labelle et al. [1] have confirmed the presence of corneal vascularization and inflammation. Later, Vallone et al. [8] utilized in vivo confocal microscopy to demonstrate the presence of inflammation and corneal vascularization. Our findings indicate a lower percentage of corneal vascularization compared to Vallone et al. [8], which supports their preference for the term “pigmentary keratitis”. The nature of the relationship between corneal vascularization and the penetration of pigment into the cornea as either a cause or a consequence of disease progression remains uncertain. Nevertheless, our analysis, along with the work of Labelle et al. [1], suggests that a higher severity of corneal pigmentation corresponds to an increased risk of corneal vascularization.

### 4.3. Predisposing Factors

#### 4.3.1. Age

The prevalence of PK and the severity of corneal pigmentation vary in published studies. In our study, the average age of our patients was 6.5 years, which is higher compared to previous studies: 4.1 years [1], 3.9 years [8], and 2.8 years [1]. This difference in age among the study groups could explain the variability in reported prevalence and severity.

Sinitsina [13] reported that ocular disorders in brachycephalic breeds typically develop between 3 and 10 years of age. The relationship between age, the prevalence of PK, and the severity of corneal pigmentation is controversial. Our findings indicate that the risk of PK increases with age, particularly in animals over 8 years old. Conversely, Vallone et al. [8] concluded that age distribution did not significantly differ between dogs with PK and dogs without PK.

#### 4.3.2. Sex and Fertility Status

Multiple studies have examined the potential association between gender and PK. Krecny et al. [17] and Vallone et al. [8] both concluded that there is no statistically significant relationship between gender and the presence of this condition. However, Labelle et al. [1] found that female Pugs are more likely to have no or mild corneal pigmentation, while male Pugs are more likely to have moderate or severe forms. Additionally, spayed female Pugs have a higher likelihood of not developing PK compared to males, suggesting that hormonal influences may play a role in its incidence and severity. Further research is needed to better understand the connection between sex, fertility status, and PK.

#### 4.3.3. Coat Color

Although there are four coat colors in the Pug breed: beige, black, silver, and apricot, according to the official breed standard, Labelle et al. [1] classified them into two groups: black and beige. This factor did not influence the presence of PK, but they described a statistically significant relationship with the severity of the disease. Specifically, they described that significantly more Pugs with a beige coat had moderate PK than Pugs with a black coat, and significantly more Pugs with a black coat had mild PK than Pugs with a beige coat. No significant differences were identified in our study when comparing color coats, which were categorized as dark coats. Furthermore, no correlation was found between the dark and light color coat groups and the severe corneal pigmentation group.

### 4.4. Tear Film Analysis

#### 4.4.1. Schirmer Tear Test (STT)

Tear production does not decrease during the early stages of the disease in Pugs, contradicting the notion that decreased tear quantity is associated with more severe and aggressive corneal pigmentation [1]. The mean STT of 16.3 ± 5.8 mm/min in both eyes found in this study is lower than previously reported values, which may be attributed to the older age and more severe corneal pigmentation of the studied Pugs. However, this decrease in STT does not affect the presence of PK. Furthermore, the results of this study support the findings of previous studies [1], as lower STT values are observed to coincide with more severe forms of PK.

#### 4.4.2. Tear Ferning Test (TFT)

The TFT is a technique utilized to assess tear quality in humans and horses. Recently, this test has been described for use in dogs [23] and was also evaluated in our current study. Our findings indicate that 38.5% of the Pugs tested had a TFT type II result. Statistical analysis revealed no significant association between TFT results and the presence of PK. However, a notable correlation was observed between TFT results and the severity of the corneal pigmentation. Tear quality was categorized into two groups based on abnormal (type III and type IV) or normal (type I and type II) characteristics. The group exhibiting the highest corneal pigmentation severity was compared to the remaining groups, revealing a statistically significant difference. Consequently, a worse TFT result was associated with a higher risk of having a more severe form of the disease (OR = 3.540, 95% CI: 1.667–7.517). The TFT is a quick, cost-effective assessment that indirectly reflects tear osmolarity and quality, which is believed to be related to mucin production [31]. Our results suggest that the TFT can serve as a valuable tool for evaluating tear quality in dogs. It offers an easy, affordable, rapid, and safe approach that can aid in assessing the initial state of the disease and facilitate improved disease monitoring.

#### 4.4.3. Tear Break-Up Time (TBUT)

In dogs, the mean TBUT test values range from 19.7 ± 5 s to 21.5 ± 7.4 s [11,27]. The high prevalence of low TBUT test values in Pugs has been consistently supported by multiple studies [1,26]. In a study conducted by Krecny et al. [17], TBUT test measurements were obtained from only five Pugs. Among these, four Pugs had an abnormal TBUT test, measuring less than 20 s in both eyes, while only one Pug had a normal TBUT test of 19 s in one eye and 21 s in the other. All these dogs exhibited corneal changes, including corneal pigmentation, opacity, ulceration, and corneal vascularization. In our own study, the TBUT test was qualitatively evaluated to assess tear production in each eye. The mean TBUT test was found to be 6.0 ± 2.7 s in both eyes combined. The results revealed a statistically significant relationship between the TBUT test and the presence of PK, with a higher mean TBUT test observed in dogs without PK. Additionally, the mean TBUT test was observed to decrease as the severity of the pathology increased, although the significance of this observation could not be determined.

Labelle et al. [1] demonstrated that Pug dogs with severe corneal pigmentation exhibit lower tear production and TBUT test values. However, Pugs with very mild, mild, and moderate corneal pigmentation do not show significantly reduced tear production or TBUT test values compared to Pug-breed dogs without PK. This suggests that decreased tear production and the TBUT test may be a consequence of chronic corneal disease rather than a cause. The mean STT value (OD: 22 mm/min; OS: 23 mm/min) observed in this study is consistent with values reported for other dog breeds. However, the TBUT test values (OD: 8 s; OS: 9 s) are lower than those reported for mesocephalic breeds and similar to those observed in brachycephalic breeds.

### 4.5. Corneal Sensitivity

Brachycephalic breeds exhibit reduced corneal sensitivity in comparison to dolichocephalic and mesocephalic breeds [18]. This diminished sensitivity makes the eye more susceptible to damage, and common reflex responses include rapid blinking, globe retraction, and nictitating membrane prolapse [32]. Labelle et al. [1] utilized corneal stensiometry to evaluate corneal sensitivity in Pug-breed dogs, and their findings aligned with previous reports on brachycephalic breeds [33]. Our study found that none of the animals displayed a positive response to fine sensitivity, with the majority exhibiting negative fine sensitivity (67.6%) and reduced sensitivity (32.4%). Regarding gross sensitivity, only 10% of the eyes showed positive sensitivity, while the majority had reduced gross sensitivity (79.9%). No significant differences were observed between corneal sensitivity and the PK presence or severity of corneal pigmentation groups. However, a statistically significant association was found between fine corneal sensitivity and maximum severity (OR = 2.108, 95% CI: 1.004–4.427), indicating that negative fine sensitivity is a risk factor for increased severity in corneal pigmentation. Furthermore, our findings suggest that advanced corneal disease is linked to reduced corneal sensitivity, although it is possible that pigment accumulation contributes to reduced sensitivity rather than the loss of sensitivity leading to pigment accumulation [34]. Further investigation is necessary to evaluate corneal sensitivity in the early stages of corneal pigmentation and its relationship with the overall outcome.

### 4.6. Corneal Thickness

Pachymetry was measured at the beginning of the study with the intention of using it as an indicator for the follow-up of treated patients, with the hypothesis that pigment removal would occur when the treatment was applied. However, we observed that the pigment was not removed homogeneously and that it was also very difficult to always measure at the same point on the cornea. According to Gilger et al. [35], the average corneal thickness in dogs, as measured by ultrasonic pachymetry, is 562 ± 62 μm. However, our results indicate that the average pachymetry was 697.9 ± 134.1 μm in the temporal zone and 762.0 ± 144.8 μm in the nasal zone. These findings demonstrate statistically significant differences in pachymetry between the nasal and temporal zones, regardless of the presence or absence of PK. Additionally, our data supports the results reported by Labelle et al. [1], which showed that the corneal thickness is greater in the nasal zone compared to the temporal zone. Furthermore, their findings suggested that pigment enters the cornea through the nasal zone and advances towards the central zone. Taken together, these findings imply that pigment not only occupies a larger corneal area but also progressively increases the thickness of the cornea as it accumulates.

## 5. Conclusions

In conclusion, our study found a high prevalence of PK in Pug-breed dogs in the Canary Islands, suggesting a potential genetic basis for the disease. The severity of corneal pigmentation was associated with age, with older dogs being at a higher risk. Female Pugs, particularly those that were spayed, had a lower likelihood of developing PK. Previous ocular diseases and alterations in eyelid conformation were also found to be potential risk factors for PK. Corneal vascularization and reduced corneal sensitivity were commonly observed in cases of PK. Tear film analysis revealed lower TFT and TBUT tests in dogs with more severe forms of PK. The TFT and corneal pachymetry were also found to be useful in evaluating tear quality and corneal thickness, respectively. Further research is needed to better understand the etiology and underlying mechanisms of PK in Pug-breed dogs.

## Figures and Tables

**Figure 1 animals-14-00580-f001:**
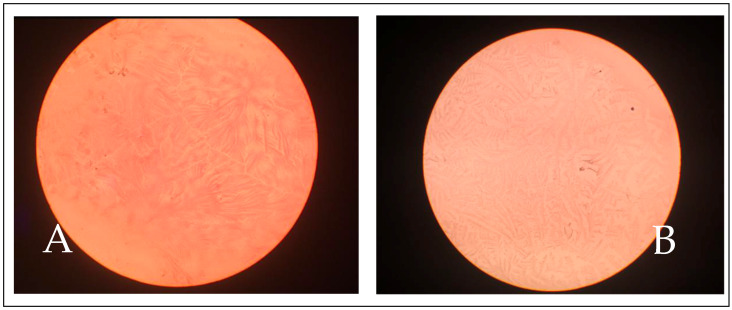
Tear Ferning Test (TFT) patterns. Examples of tear fern grade I (**A**), tear fern grade II (**B**), tear fern grade III (**C**), and tear fern grade IV (**D**).

**Figure 2 animals-14-00580-f002:**
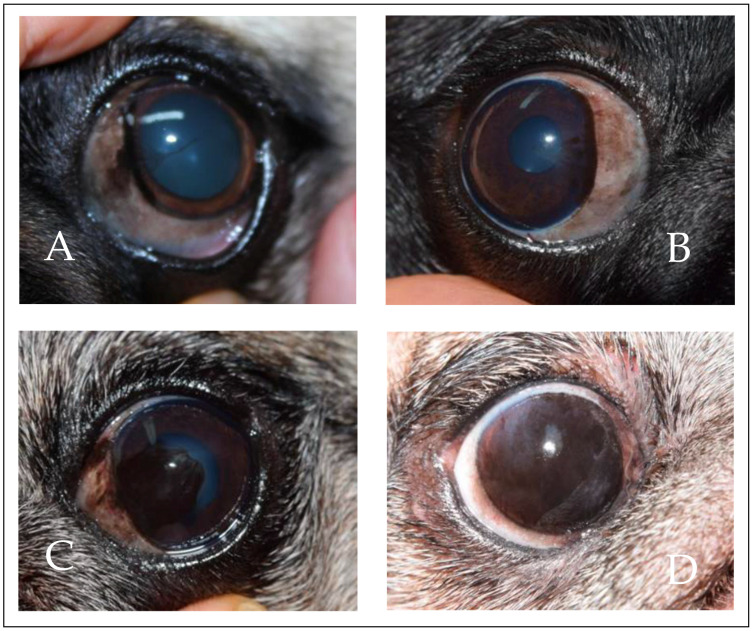
Severity of corneal pigmentation. Examples of very mild (**A**), mild (**B**), moderate (**C**), and severe (**D**).

**Table 1 animals-14-00580-t001:** Description of previous ocular diseases and clinical symptoms in the 207 Pug eyes with corneal pigmentation included in the study (N = number of eyes).

	Presence	Absence	*p* *
Previous ocular disease *According to owners*:	94 (42.9%)	113 (57.1%)	0.014
*Corneal ulcers*	48	-	
*Conjunctivitis*	40	-	
*Keratoconjunctivitis sicca*	6	-	
Euryblepharon	175 (84.6%)	32 (15.4%)	0.366
Trichiasis of the nasal fold	15 (7.2%)	192 (92.8%)	0.482
Trichiasis of the caruncle	175 (84.6%)	32 (15.4%)	0.366
Distiquiasis	26 (12.5%)	181 (87.5%)	0.026
Upper nasal entropion	33 (15.5%)	174 (85.5%)	0.396
Inferior nasal entropion	205 (9.99%)	2 (0.01%)	0.067
Corneal vascularization	120 (57.9%)	87 (42.1%)	0.001

* The *p*-value is shown for the comparison between the different severity groups of corneal pigmentation.

**Table 2 animals-14-00580-t002:** Relationship between the presence of PK and predisposing factors.

	Total of Animals (N = 110)	Eyes with PK (N = 207)	Eyes without PK (N = 12)	Total Eyes (N = 209)	*p*
Age	0–3 years (N = 12)	17	7	24	0.001
	4–7 years (= 56)	109	3	112	
	>8 years (N = 42)	81	2	83	
Sex	Female (N = 51)	94	7	101	0.762
	Male (N = 59)	113	5	118	
Neuter status	Not Neutered (N = 77)	144	9	153	0.159
	Neutered (N = 33)	63	3	66	
Coat color	Dark (N = 16)	30	2	32	0.836
	Light (N = 94)	177	10	187	

**Table 3 animals-14-00580-t003:** Relationship between the severity of corneal pigmentation according to the classification of Labelle et al. [1] and the predisposing factors in Pug dogs with PK.

		Severity of Corneal Pigmentation				Total	*p*
		Very Mild	Mild	Moderate	Severe		
Age	0–3 years (N)	6	4	7	0	17	0.001
	4–7 years (N)	29	21	41	18	109	
	>8 years (N)	7	9	31	34	81	
Sex	Female	19	13	36	26	94	0.765
	Male	23	21	43	26	113	
Neuter status	Not Neutered	31	29	51	33	144	0.103
	Neutered	11	5	28	19	63	
Coat color	Dark	14	6	5	5	30	0.001
	Light	28	28	74	47	177	
Corneal vascularization	Presence	7	7	58	48	120	0.001
	Absence	35	27	21	4	87	

**Table 4 animals-14-00580-t004:** Relationship between the presence of pigmentary keratitis and the results of an ophthalmologic examination.

	Units/N	Pigmentary Keratitis (N = 207)	No Pigmentary Keratitis (N = 12)	Total (N = 219)	*p*
Schirmer Tear Test	mm/min	16.1 ± 5.8	18.9 ± 3.8	16.3 ± 5.8	0.102
Tear Ferning Test (TFT)	Type I (N)	21	2	23	0.408
	Type II (N)	65	4	69	
	Type III (N)	51	2	53	
	Type IV (N)	34	0	34	
Tear Break-Up Time (TBUT)	Seconds	5.9 ± 2.7	7.6 ± 2.2	6.0 ± 2.7	0.038
Fine corneal sensitivity	Negative (N)	140	8	148	0.005
	Reduced (N)	67	4	71	
Gross corneal sensitivity	Negative (N)	22	0	22	0.397
	Reduced (N)	165	10	175	
	Positive (N)	20	2	22	
Corneal thickness	Nasal (µm)	770.0 ± 143.6	608.2 ± 55.1	762.0 ± 144.8	0.001
	Temporal (µm)	702.6 ± 135.7	605.4 ± 33.6	697.94 ± 134.1	0.014

**Table 5 animals-14-00580-t005:** Relationship between the severity of corneal pigmentation according to the classification of Labelle et al. [1] and the results of ophthalmologic examination in Pug dogs with PK.

		Severity of Corneal Pigmentation				Total	*p*
		Very Mild	Mild	Moderate	Severe		
Schirmer Tear Test	mm/min	19.8 ± 4.3	19.2 ± 4.7	16.7 ± 4.4	10.1 ± 5.1	16.1 ± 5.8	0.001
Tear Ferning Test (TFT)	Type I (N)	7	4	4	6	21	0.002
	Type II (N)	17	13	29	6	65	
	Type III (N)	10	7	21	13	51	
	Type IV (N)	4	4	8	18	34	
Tear Break-Up Time (TBUT) test	Seconds (s)	6.9 ± 3.6	6.4 ± 2.9	6.0 ± 2.1	4.6 ± 2.3	5.9 ± 2.7	0.001
Fine corneal sensitivity	Negative (N)	23	24	52	41	140	0.093
	Reduced (N)	19	10	27	11	67	
Gross corneal sensitivity	Negative (N)	2	5	7	8	22	0.218
	Reduced (N)	38	27	60	40	165	
	Positive (N)	2	2	12	4	20	
Corneal thickness	Nasal (µm)	673.0 ± 79.5	658.2 ± 92.3	766.1 ± 102.0	927.9 ± 117.9		0.001
	Temporal (µm)	651.6 ± 64.2	635.6 ± 62.5	642.2 ± 55.6	871.7 ± 149.7		0.001

## Data Availability

The data that support the findings of this study are available from the corresponding author upon reasonable request.

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
