# Peer review of "Ocular Surface Characteristics in Pugs with Pigmentary Keratitis in the Canary Islands, Spain"

_animals, 2024, doi:10.3390/ani14040580_

Round 1
Reviewer 1 Report
Comments and Suggestions for Authors
This is a valuable epidemiological study of a large number of cases in a geographically isolated location on an island. Unfortunately, the submitted paper has many problems for a scientific paper. The current paper does not allow for a good understanding by the reader. There are major problems in the way the results are organized and we have pointed out a large number of them. The paper needs major improvements, but it should definitely be published in our journal. Please reorganize the results and improve the discussion as well. All the reviewers and editors would like to work together sincerely towards the successful acceptance of this article.
> Throughout the paper.
The confusion between risk factors for the onset of disease and symptoms makes it difficult to read. Each item detected needs to be reorganised; it is difficult to classify and discuss neovascularisation as an incident risk factor in this study, where 95% were classified in the incident group.
Literature citation methods must be standardised. There is a mixture of references cited by reference number and those cited by author name and year of publication only. The submission rules must be complied with, and where authors are cited by name and year of publication, they must also be numbered.
It is sufficient to indicate up to one decimal place.
>> Title
"Canary Islands, Spain" should be included. "without PK" should not be included. Because 95% of cases developed PK.
>> For the entire RESULTS.
Table 1 and Table 4 are paired、and also Table 2 and Table 3 respectively. Table 4 should be moved after Table 1 and the text should be in that order.
Separating the comparison between each element and the presence or absence of PK and the comparison between each element and the severity of PK respectively prevents good understanding. They should be presented in the same paragraph or in the
consecutive paragraphs.
Corresponding table numbers must be clearly stated in the text.
> Paragraphs 2 184~191.
The relationship between “history of ocular diseases” and the presence or absence of PK should be shown in a table.
> Paragraphs 3 184~200, paragraph 15 294~297, paragraph 16 298~302
It is difficult to understand why the table does not show the association with symptoms.
If these paragraphs are "preexisting ocular condition" paragraphs, it would be better to put the three paragraphs together.
If "Vascularisation" and "Corneal vessels" are the same, it is better to unify the terms for good understanding. In the ocular manifestations, the reason for focusing only on "Corneal vessels" should be provided.
It needs to be clarified whether it is a "predisposing factor" or a "clinical feature".
> Paragraphs 4, 201~208 and 5, 201~210.
Each of these items is given in more detail later in the document. It would be better to delete them and state them in their own paragraphs.
"Reduced" or "decreased" should be unified in the evaluation of "sensitivity".
"Positive" in "Gross corneal sensitivity" should be clearly indicated in Table 2.
> Table 1.
"Total" should include the number of head as well as the number of eyes.
"Neuter status" should be presented separately for males and females.
"Corneal vessels" should also be presented.
> Table 2.
Please also indicate "Positive" for "Gross corneal sensitivity".
> Tables 3 and 4.
"Ausencia" must be presented with an additional column.
> Paragraphs 6 214~219, 13 273~281.
It is better to organise the content and put them in the same paragraph or make them consecutive paragraphs.
Is this a paragraph corresponding to Table 1? The content of the text is not organised in Table 1.
The comparison of "two age groups" interrupts the flow of the paper.
> Paragraphs 7 220~224, 14 286~293.
It is better to organise the contents and put them in the same paragraph or in consecutive paragraphs.
> Paragraphs 8 225~233, 10 248~252, 11 253~259.
It is better to organise the contents and put them in the same paragraph or in consecutive paragraphs.
> Paragraphs 9 234~242, 12 260~267.
It would be better to organise the contents and put them in the same paragraph or make them into consecutive paragraphs.
The logical development is uncomfortable. The logical structure should be reconsidered.
>> Discussion
I seek significant improvements in the structure and logic of the RESULTS. I expect that there will be major improvements in DISCUSSION as a result of the changes in RESULTS. Therefore, you must limit yourself to major comments only.
Comparisons with previous epidemiological studies should be organised using tables. It can be emphasised that the frequency of the disease is higher in the Canary Islands and that genetic background is therefore strongly involved. Climatic and endemic differences can be discussed.
In this study, where many disease groups are represented, it is difficult to discuss the various symptoms and the test results as risk factors for the onset of the disease. I strongly feel that the DISCUSSION should be developed based on the significantly changed RESULTS.
The value of presenting this study is very high.
I trust you will do your best, even if it takes a great deal of time and effort to improve it.
Author Response
Thank you very much for all your contributions, we believe that the manuscript has now been substantially improved thanks to your help. We are very grateful to you.
> Throughout the paper.
The confusion between risk factors for the onset of disease and symptoms makes it difficult to read. Each item detected needs to be reorganised; it is difficult to classify and discuss neovascularisation as an incident risk factor in this study, where 95% were classified in the incident group.
The entire manuscript has been reorganized, separating all the results by the different sections and following the same order in the discussion. Indeed, we had everything a bit mixed up and now it has been better clarified.
Some sentences have been eliminated, as a result of the completed bibliographic review, which did not contribute anything to our study.
Literature citation methods must be standardised. There is a mixture of references cited by reference number and those cited by author name and year of publication only. The submission rules must be complied with, and where authors are cited by name and year of publication, they must also be numbered.
A complete revision of the citation has been completed.
It is sufficient to indicate up to one decimal place
Deleted the second decimal throughout the paper. As a consequence, some numbers have had to be rounded and the text and content of the tables have been modified.
>> Title
"Canary Islands, Spain" should be included. "without PK" should not be included. Because 95% of cases developed PK.
Done!
>> For the entire RESULTS.
Table 1 and Table 4 are paired、and also Table 2 and Table 3 respectively. Table 4 should be moved after Table 1 and the text should be in that order.
Separating the comparison between each element and the presence or absence of PK and the comparison between each element and the severity of PK respectively prevents good understanding. They should be presented in the same paragraph or in the consecutive paragraphs.
Corresponding table numbers must be clearly stated in the text.
Done!
> Paragraphs 2 184~191.
The relationship between “history of ocular diseases” and the presence or absence of PK should be shown in a table.
We have included a new table (Table 1) with the Description of previous ocular and clinical symptoms in the 207 Pug eyes with corneal pigmentation included in the study. The p-value is shown for the comparison between the different severity groups of corneal pigmentation.
> Paragraphs 3 184~200, paragraph 15 294~297, paragraph 16 298~302
It is difficult to understand why the table does not show the association with symptoms.
If these paragraphs are "preexisting ocular condition" paragraphs, it would be better to put the three paragraphs together.
If "Vascularisation" and "Corneal vessels" are the same, it is better to unify the terms for good understanding. In the ocular manifestations, the reason for focusing only on "Corneal vessels" should be provided.
It needs to be clarified whether it is a "predisposing factor" or a "clinical feature".
Done!
> Paragraphs 4, 201~208 and 5, 201~210.
Each of these items is given in more detail later in the document. It would be better to delete them and state them in their own paragraphs.
"Reduced" or "decreased" should be unified in the evaluation of "sensitivity".
"Positive" in "Gross corneal sensitivity" should be clearly indicated in Table 2.
Done!
> Table 1.
"Total" should include the number of head as well as the number of eyes.
Done!
"Neuter status" should be presented separately for males and females.
Done!
"Corneal vessels" should also be presented.
Done!, but included in Table 1, not in table 2, because it is a clinical symptoms and not a predisposing actor.
> Table 2.
Please also indicate "Positive" for "Gross corneal sensitivity".
Done!
> Tables 3 and 4.
"Ausencia" must be presented with an additional column.
Information provided in table 4 (new number)
> Paragraphs 6 214~219, 13 273~281.
It is better to organise the content and put them in the same paragraph or make them consecutive paragraphs.
Is this a paragraph corresponding to Table 1? The content of the text is not organised in Table 1.
The entire manuscript has been reorganized,
The comparison of "two age groups" interrupts the flow of the paper.
Eliminated
> Paragraphs 7 220~224, 14 286~293.
It is better to organise the contents and put them in the same paragraph or in consecutive paragraphs.
The entire manuscript has been reorganized,
> Paragraphs 8 225~233, 10 248~252, 11 253~259.
It is better to organise the contents and put them in the same paragraph or in consecutive paragraphs.
The entire manuscript has been reorganized,
> Paragraphs 9 234~242, 12 260~267.
It would be better to organise the contents and put them in the same paragraph or make them into consecutive paragraphs.
The logical development is uncomfortable. The logical structure should be reconsidered.
The entire manuscript has been reorganized,
>> Discussion
I seek significant improvements in the structure and logic of the RESULTS. I expect that there will be major improvements in DISCUSSION as a result of the changes in RESULTS. Therefore, you must limit yourself to major comments only.
The entire manuscript has been reorganized and discussion has been revised.
Comparisons with previous epidemiological studies should be organised using tables.
A comparison of epidemiological results is not the aim of this study. Therefore, we have included only several examples of prevalences form other latitudes. We consider that a table in discussion is not appropriate under our point of view.
It can be emphasised that the frequency of the disease is higher in the Canary Islands and that genetic background is therefore strongly involved. Climatic and endemic differences can be discussed.
Done!
In this study, where many disease groups are represented, it is difficult to discuss the various symptoms and the test results as risk factors for the onset of the disease. I strongly feel that the DISCUSSION should be developed based on the significantly changed RESULTS.
We have reduced the discussion, and several sentences not related to our results has been deleted.
The value of presenting this study is very high.
I trust you will do your best, even if it takes a great deal of time and effort to improve it.
Thanks for your help
Reviewer 2 Report
Comments and Suggestions for Authors
The study is an examination of pigmentary keratitis in pug dogs in the Canary Islands. The authors assess the prevalence and severity of pigmentary keratitis in these dogs and also correlate these findings with other variables including clinical history, diagnostic findings, and signalment.
Overall, this paper is scientifically sound. The examination of this ocular pathology in pugs is thorough in its clinical and diagnostic correlates, and the statistical tests are appropriate for these analyses. I have no edits regarding the content or organization of the paper.
Comments on the Quality of English LanguageThere are few typos or grammatical errors that may be corrected with minimal editing:
Line 28: “describe” should be “describes”
Line 270-271: “Pugs dogs” should be “Pug dogs”
Line 397: “brachiocephalic dogs” should be “brachycephalic dogs”
Line 463: “brachiocephalic dogs” should be “brachycephalic dogs”
Line 468: “pimgentation” should be “pigmentation”
Capitalization of terms should also be more consistent. For example, pigmentary keratitis is capitalized in some lines (e.g., line 72) but not others (e.g., line 193).
Author Response
Thank you very much for all your contributions, we believe that the manuscript has now been substantially improved thanks to your help. We are very grateful to you.
Comments on the Quality of English Language
There are few typos or grammatical errors that may be corrected with minimal editing:
Line 28: “describe” should be “describes”
corrected
Line 270-271: “Pugs dogs” should be “Pug dogs”
corrected
Line 397: “brachiocephalic dogs” should be “brachycephalic dogs”
corrected
Line 463: “brachiocephalic dogs” should be “brachycephalic dogs”
corrected
Line 468: “pimgentation” should be “pigmentation”
corrected
Capitalization of terms should also be more consistent. For example, pigmentary keratitis is capitalized in some lines (e.g., line 72) but not others (e.g., line 193).
Pigmentary Keratitis has been spelled correctly in capitals and PK throughout the text.
Reviewer 3 Report
Comments and Suggestions for Authors
Please find comments below.
Introduction:
11. To draw more attention to readers, further information regarding complications of PK should be addressed.
Materials and Methods
11. From tear film analysis section, STT was performed first and then followed by TFBUT, flu staining, TFT, corneal sensitivity, and pachymetry. According to the sequence of method, is there any concerns/impacts to the results?
22. Figure 1A: It is convincing if it is a tear fern grade I. Please consider selecting an appropriate representative of grade I.
33. It is suggested to replace the words “lateral” and “medial” with “temporal” and “nasal”.
44. A more specific area of pachymetry should be described. If that particular area for pachymetry is occupied with pigment, would the area of measurement be changed?
Discussion
11. Do you do think geography influences your quantitative and/or qualitative tear values? If so, please add comments accordingly.
22. What is the possible cause of increased corneal thickness in the nasal area, compared to the temporal?
Author Response
Thank you very much for all your contributions, we believe that the manuscript has now been substantially improved thanks to your help. We are very grateful to you.
Introduction:
- To draw more attention to readers, further information regarding complications of PK should be addressed.
Included reference to visual impairment and blindness in severe cases.
Materials and Methods
- From tear film analysis section, STT was performed first and then followed by TFBUT, flu staining, TFT, corneal sensitivity, and pachymetry. According to the sequence of method, is there any concerns/impacts to the results?
The entire manuscript has been reorganized, separating all the results by the different sections and following the same order in the discussion. Indeed, we had everything a bit mixed up and now it has been better clarified.
- Figure 1A: It is convincing if it is a tear fern grade I. Please consider selecting an appropriate representative of grade I.
We have reviewed the image and it is consistent with a type I.
- It is suggested to replace the words “lateral” and “medial” with “temporal” and “nasal”.
corrected
- A more specific area of pachymetry should be described. If that particular area for pachymetry is occupied with pigment, would the area of measurement be changed?
A sentence for clarification has been added in the discussion (at the beginning of 4.6 Corneal Thickness.
Discussion
- Do you do think geography influences your quantitative and/or qualitative tear values? If so, please add comments accordingly.
We have added a sentence at the end of discussion 4.1 Prevalence of PK, related to this possibility.
- What is the possible cause of increased corneal thickness in the nasal area, compared to the temporal?
Because more pigment is deposited in the nasal area as a consequence of other ocular diseases (distiquiasis, triquiasis, entropion, etc) as describe in Table 1 and predisposing factors of the disease that promote the deposition of pigment more in that area.
However, the PK is defined by a pigmentation that is originated from the nasal aspect of the corneal limbus. Several theories has been developed related the origin of the pigment, but this is not the aim of this manuscript).
Thanks for your help
Round 2
Reviewer 1 Report
Comments and Suggestions for Authors
The revisions were made sincerely in accordance with the reviewer's comments.
The organization of paragraphs with titles has consolidated duplicated content.
It became very easy to read.
It should now be accepted as a scientific paper.
However, there are a few places where the data is displayed to two decimal places.
For example, line418~423.
Please correct these.